

# Ginsenoside Rk1 bioactivity: a systematic review

Abdelrahman Elshafay[1], Ngo Xuan Tinh[2], Samar Salman[3], Yara Saber Shaheen[4], Eman Bashir Othman[5], Mohamed Tamer Elhady[6], Aswin Ratna Kansakar[7], Linh Tran[8], Le Van[2], Kenji Hirayama[9] and Nguyen Tien Huy[10,11]

[1] Faculty of Medicine, Al-Azhar University, Cairo, Egypt
[2] Faculty of Pharmacy, University of Medicine and Pharmacy, Ho Chi Minh city, Vietnam
[3] Tanta University Hospital, Tanta, Egypt
[4] Faculty of Medicine, Cairo University, Cairo, Egypt
[5] Department of Medicine, Tripoli Central Hospital, Tripoli, Libya
[6] Department of Pediatrics, Zagazig University Hospitals, Sharkia, Egypt
[7] Dirghayu Guru Hospital and Research Center, Kathmandu, Nepal
[8] Institute of Research and Development, Duy Tan University, Da Nang, Vietnam
[9] Department of Immunogenetics, Institute of Tropical Medicine (NEKKEN), Graduate School of Biomedical Sciences, Nagasaki University, Nagasaki, Japan
[10] Evidence Based Medicine Research Group & Faculty of Applied Sciences, Ton Duc Thang University, Ho Chi Minh City, Vietnam
[11] Department of Clinical Product Development, Institute of Tropical Medicine (NEKKEN), Leading Graduate School Program, and Graduate School of Biomedical Sciences, Nagasaki University, Nagasaki, Japan

Corresponding author
Nguyen Tien Huy,
nguyentienhuy@tdt.edu.vn

## ABSTRACT

Ginsenoside Rk1 (G-Rk1) is a unique component created by processing the ginseng plant (mainly Sung Ginseng (SG)) at high temperatures. The aim of our study was to systematically review the pharmacological effects of G-Rk1. We utilized and manually searched eight databases to select *in vivo* and *in vitro* original studies that provided information about biological, pharmaceutical effects of G-Rk1 and were published up to July 2017 with no restriction on language or study design. Out of the 156 papers identified, we retrieved 28 eligible papers in the first skimming phase of research. Several articles largely described the G-Rk1 anti-cancer activity investigating "cell viability", "cell proliferation inhibition", "apoptotic activity", and "effects of G-Rk1 on G1 phase and autophagy in tumor cells" either alone or in combination with G-Rg5. Others proved that it has antiplatelet aggregation activities, anti-inflammatory effects, anti-insulin resistance, nephroprotective effect, antimicrobial effect, cognitive function enhancement, lipid accumulation reduction and prevents osteoporosis. In conclusion, G-Rk1 has a significant anti-tumor effect on liver cancer, melanoma, lung cancer, cervical cancer, colon cancer, pancreatic cancer, gastric cancer, and breast adenocarcinoma against *in vitro* cell lines. *In vivo* experiments are further warranted to confirm these effects.

## INTRODUCTION

Ginseng is commonly known as a medicinal herb that is obtained from the roots of genus *Panax* (*Shin, Kwon & Park, 2015*). Ginseng belongs to one of the most ancient herbs

in traditional medicine and is still widely used today (*Choi et al., 2013*). Ginsenosides are classified based on the steroidal structure and the number of hydroxyl groups/sugar moieties attached to it, such as protopanaxadiol, protopanaxatriol, oleanolic acid (or aglycone oleanolic acid) and ocotillol (*Nag et al., 2015*). The protopanaxadiol group includes Rb1, Rb2, Rb3, Rc, Rd, Rg3, Rh2, Rs1, and Rk1. The protopanaxatriol group includes Re, Rf, Rg1, Rg2, and Rh1 (*Kim, Kim & Shin, 2013*). Ro is classified as an oleanolic acid group (*Tachikawa et al., 1999*). Details of types of ginsenosides are presented in Fig. 1.

The quality and composition of ginsenosides in the ginseng plant are affected by a range of factors such as species, age, part of the plant itself, method of cultivation, harvesting season and preservation methods (*Lim, Mudge & Vermeylen, 2005*; *Schlag & McIntosh, 2006*). Some of the ginsenosides, e.g., Rk1, Rg3, Rg5, F4, are isolated from the heat-processed ginseng, Sun ginseng (SG), but are not detected in raw or air-dried ginseng (*Kim et al., 2000*).

Ginsenosides are widely known to have many pharmacological activities (*Choi, 2008*; *Ernst, 2010*) such as anti-tumor, anti-inflammatory (*Chen et al., 2007*), anti-fatigue (*Tang et al., 2008*) and analgesic effects (*Nemmani & Ramarao, 2003*).

Ginseng plant is commonly harvested after four to six years of cultivation and is divided in three types based on the processing methods: (1) fresh ginseng which is less than four years old, (2) white ginseng from four to six years and is oven dried after peeling, (3) red ginseng which is six years and steamed before drying. These processing methods aim to improve the efficacy, safety, and preservation (*Yun, 2001*). SG was recently developed by heat-treatment at high temperature and pressure, which were higher than those applied to the conventional preparation of red ginseng.

SG has shown higher concentrations of less polar ginsenosides, which were either entirely absent or present in trace amounts in conventional red ginseng (*Keum et al., 2000*; *Kwon et al., 2001*).

The ginsenoside Rk1 (G-Rk1) is one of the main elements of SG (*Kim et al., 2008*). Various studies confirmed the anti-cancer effects of G-Rk1 on several neoplastic such as hepatocellular carcinoma and melanoma (*Kim et al., 2012*; *Kim et al., 2008*). In recent studies, G-Rk1 was confirmed as a new endothelial barrier enhancer, which is capable of preventing or even blocking the vascular endothelial growth factor (VEGF)-induced vasopermeability in the endothelial cells. This presents the potential of developing pharmaceuticals that may effectively control pathologic vascular leakages (*Maeng et al., 2013*). Therefore, we aimed to systematically review the bioactivities of G-Rk1 in both human and animals.

## METHODS

### Protocol and registration

The Preferred Reporting Items for Systematic Reviews and Meta-Analysis (PRISMA) Checklist (*Moher et al., 2009*) was followed in this systematic review. Our protocol was registered at PROSPERO CRD42016029129 in January 2016.

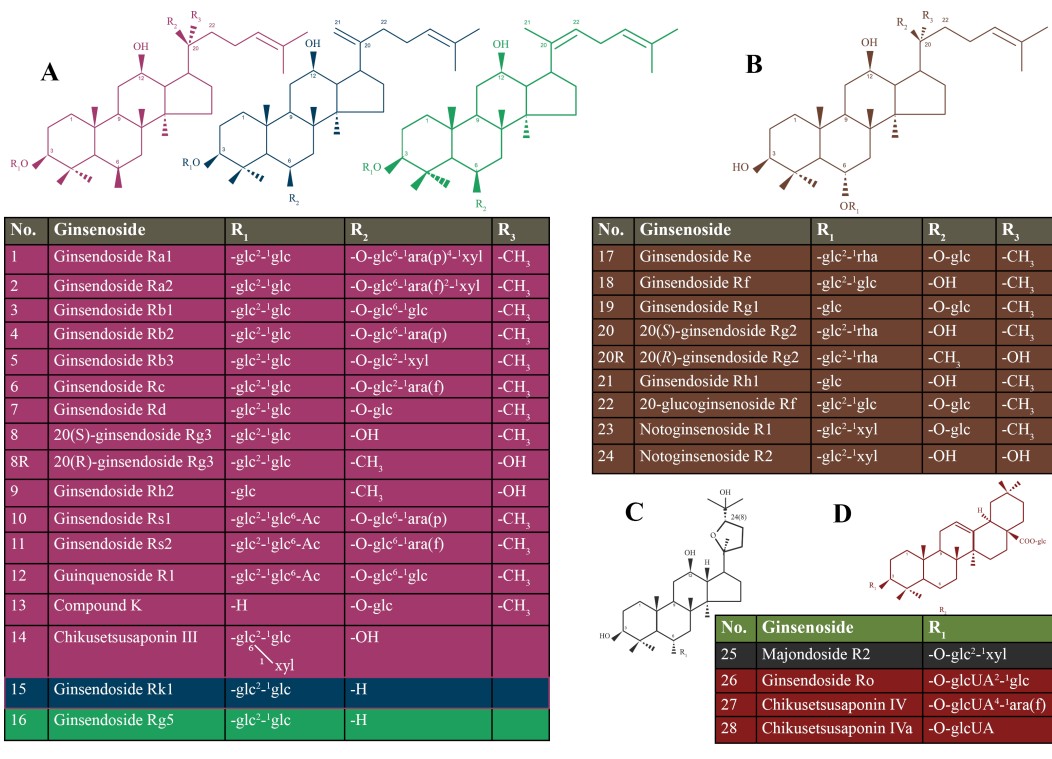

**Figure 1 Chemical structure of the ginsenosides types.** (A) protopanaxadiol (PPD)-type ginsneosides including Rk1 represented in blue color, Rg5 represented in green color, and the rest of PPD-types are in violet; (B) protopanaxatriol (PPT)-type ginsneosides represented in brown color; (C) Ocotillol- type gin- sneoside is represented in gray color; (D) Oleanic acid-type ginsneosides are represented in red color. glc, b-D-glucose; rha, a-L-rhamnose; arap, a-L-arabinose (pyranose); araf, a-L-arabinose (furanose).

## Eligibility criteria

We selected only original studies published up to July 2017 that provided information about the biological and pharmaceutical effects of G-Rk1. We included articles with G-Rk1 biological effects on human and animals either *in vivo* or *in vitro* with no restriction regarding publication language, publication date, or study design.

We excluded three main types of studies which are: (1) Studies with unreliable extracted data or overlapping data set; (2) studies with only abstract available or no full-text available; (3) books, reviews, meta-analysis studies, conference papers, and thesis. Any disagreement was discussed carefully among three reviewers to get a final decision.

## Information sources and search strategies

We conducted electronic searches using eight databases which include: PubMed, Scopus, ISI Web of Science, Google Scholar, SIGLE (System for Information on Grey Literature in Europe), Virtual Health Library (VHL), World Health Organization Global Health Library (GHL), and POPLINE. A Manual search using reference lists of studies was performed to find more relevant studies. The search strategy was performed by (AE, NXT, SS, YSS, EBO, MTE, ARK) and more information on search strategy was provided in Table S1.

## Study selection

We selected articles in two phases: (1) Title and abstract screening of all searched articles; (2) full-text screening. The articles which were not in agreement with our inclusion and exclusion criteria were excluded. Three independent reviewers completed these two selecting phases. When disagreement occurred, a consensus decision was made following a discussion with supervisor (NTH).

## Data collection process and data items

We prepared our primary extraction form, extracted three papers with it one by one, modified our form after each paper extraction and finally developed the extraction sheet that we used in the remaining articles. Three independent reviewers extracted the data from each paper. A discussion among the three reviewers was held to reach a consensus whenever there was a disagreement in any information retrieved. If three reviewers could not come to an agreement, the supervisor (NTH) was consulted.

   The extracted data items included the last name of the first author, year of publication, year of subject recruitment, journal name, study design, country and city of origin of cell lines, the name of the plant, and method of extraction of our targeted material (G-Rk1). If the study included animals, we extracted their species, sex, age, and weight. If it had been done *in vivo*, we extracted the name of the cell line, its origin, the main medium used in terms of either primary (isolated by authors) or commercial cell lines. Also, we extracted the name of the measured parameter, an assay for its measurement, time effect, administration time, active substance name, its concentration, mean, standard deviation, standard error, a *P* value of results and the statistical test. When the data was presented as graphs, we used Web blot digitizer software, and the average of the results from three reviewers was calculated to obtain one result.

## Risk of bias in individual studies

Two independent reviewers assessed all of the selected studies according to the GRADE method (*Guyatt et al., 2011*) to judge the quality of evidence, and any disagreement was resolved by discussion between them. Items such as limitation, inconsistency, indirectness, imprecision, publication bias, and moderate or large effect size were to be scored as "1" if there is no serious limitation or "0" if there is a serious limitation that has been defined according to GRADE criteria. Then the overall quality was to be scored as "high", "moderate", "low", or "very low" quality, according to their analysis of each study. The supervisor (NTH) was consulted when a disagreement occurs.

## Summary measures

Inhibition of cell proliferation, apoptosis, and regulation of protein expression were the main evaluated outcomes.

# RESULTS

## Study selection

We identified 317 citations using the search strategy. From these, we included 156 articles after removing the duplicates. After that, we examined the title, abstract and further

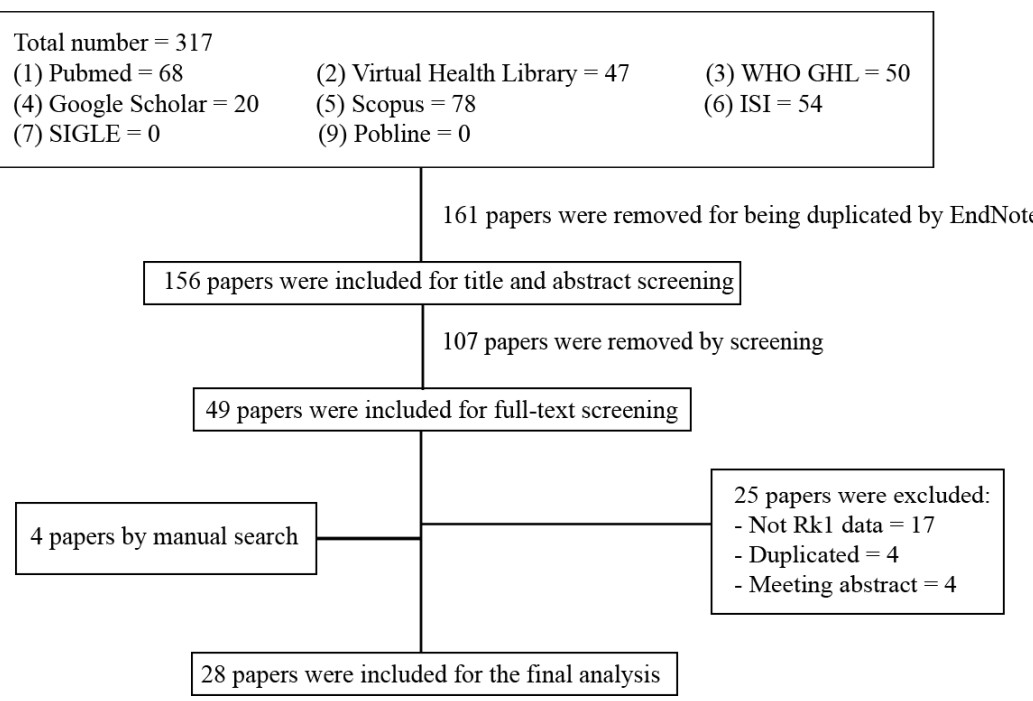

Total number = 317
(1) Pubmed = 68          (2) Virtual Health Library = 47          (3) WHO GHL = 50
(4) Google Scholar = 20  (5) Scopus = 78                          (6) ISI = 54
(7) SIGLE = 0            (9) Pobline = 0

161 papers were removed for being duplicated by EndNote

156 papers were included for title and abstract screening

107 papers were removed by screening

49 papers were included for full-text screening

25 papers were excluded:
- Not Rk1 data = 17
- Duplicated = 4
- Meeting abstract = 4

4 papers by manual search

28 papers were included for the final analysis

**Figure 2 Flowchart of our systematic review - summary of how the systematic search was conducted and eligible studies were identified (PRISMA flow diagram).** PRISMA, Preferred Reporting Items for Systematic reviews and Meta-Analyses.

excluded 107 articles. We retrieved and evaluated the full-text of the remaining 49 articles, of which 25 articles were excluded, leaving 24 articles that were eligible, in addition to four articles that were retrieved from manually searching the included references. A flowchart described in details the process of identification, inclusion, and exclusion of articles was presented in Fig. 2.

## Study characteristics

Out of the 28 studies included, 21 studies were related to the effectiveness of G-Rk1 only and seven studies were reported on the combined effects of G-Rk1 and G-Rg5. The most common study design was *in vitro* study with 22 studies (*Ahn et al., 2016*; *Ju et al., 2012*; *Kang et al., 2007*; *Kang et al., 2006*; *Kim et al., 2012*; *Kim et al., 2009*; *Kim et al., 2008*; *Kim et al., 2013b*; *Ko et al., 2009*; *Kwak & Pyo, 2016*; *Lee et al., 2009*; *Lee et al., 2010*; *Lee, 2014*; *Lim et al., 2009*; *Liu et al., 2007*; *Park et al., 2002*; *Ponnuraj et al., 2014*; *Quan et al., 2015*; *Ryu et al., 2016*; *Siddiqi et al., 2014*; *Toh et al., 2011*; *Xue et al., 2017*), while *in vivo* study was less common with only two studies (*Jing et al., 2006*; *Kim et al., 2010*). The remaining four articles were both *in vitro* and *in vivo* studies (*Bao et al., 2005*; *Hu et al., 2017*; *Maeng et al., 2013*; *Park et al., 2015*). A summary of the included studies was presented in Table 1. For G-Rk1, bioactivities and mechanism of actions were summarized in Fig. 3.

**Table 1** Study characteristics of included articles.

| Author and year | Country | Study design | Cell lines | Parameter assessed |
| --- | --- | --- | --- | --- |
| Ko et al. (2009) | South Korea | In vitro | HepG2 | Cell viability, cell proliferation, inhibitory activity (IC50) |
| Lee (2014) | South Korea | In vitro | HepG2 | Cell viability |
| Toh et al. (2011) | Singapore | In vitro | SNU449 (CRL-2234), SNU182 (CRL-2235) and HepG2 (HB-8065) | Cell viability, cell proliferation |
| Kim et al. (2008) | South Korea | In vitro | HepG2 | Cell viability, telomerase activity |
| Park et al. (2002) | South Korea | In vitro | SK-Hep-1 cells | Cell viability |
| Lim et al. (2009) | South Korea | In vitro | Junctional proteins (zo-1, occludin and plakoglobin) | ND |
| Kim et al. (2009) | South Korea | In vitro | 3T3-L1 fibroblast cells | Cell viability, lipid accumulation |
| Kim et al. (2012) | South Korea | In vitro | SK-MEL-2 human melanoma | Cell viability |
| Ju et al. (2012) | South Korea | In vitro | Platelet | Antiplatelet aggregation activity |
| Liu et al. (2007) | France | In vitro | Embryonic neural stem cells (neurospheres) | Neurogenic activity |
| Lee et al. (2009) | South Korea | In vitro | Platelet | Collagen (3–4 μg/L) induced platelet aggregation |
| Kim et al. (2010) | South Korea | In vivo | ND | TPA-induced mouse ear edema |
| Maeng et al. (2013) | South Korea | In vitro and In vivo | HREC cells | VEGF-induced retinal endothelial permeability, VEGF-induced destabilization of TJ protein ZO-1, ZO-2 and occludin in membrane and cytosol |
| Kang et al. (2007) | Japan | In vitro | ND | The OH scavenging inhibition |
| Kang et al. (2006) | Japan | In vitro | ND | The OH scavenging activities |
| Lee et al. (2010) | South Korea | In vitro | HUVECs | Cell viability |
| Kim et al. (2013a) | South Korea | In vitro | Gastric cancer AGS cell | Cell viability, the anticancer activity of ginsenosides after heat processing (IC50) |
| Bao et al. (2005) | South Korea | In vitro and In vivo | Cortical cell cultures containing neuronal and non-neuronal cells | Cognitive performance, excitotoxicity induced by NMDA and glutamate |
| Park et al. (2015) | South Korea | In vitro and In vivo | LLC-PK1 cells | Cell viability |
| Siddiqi et al. (2014) | South Korea | In vitro | The murine cell line, MC3T3-E1 | Cell viability, mineralization, ALP activity, collagen and glutathione |
| Jing et al. (2006) | China | In vivo | ND | Cognitive performance |
| Ponnuraj et al. (2014) | South Korea | In vitro | 3T3-L1 cells | Cell viability, glucose utilization |
| Ahn et al. (2016) | South Korea | In vitro | HaCaT/RAW 264.7 | Anti-inflammation activity |
| Hu et al. (2017) | China | In vivo | ND | Anti-inflammation activity |
| Kwak & Pyo (2016) | South Korea | In vitro | A549 cell | Cell viability |
| Quan et al. (2015) | China | In vitro | A549, HCT-116, HepG2, Hela, MCF-7, and PANC-1 cells | Cell viability |
| Xue et al. (2017) | China | In vitro | ND | Antimicrobial activity |
| Ryu et al. (2016) | South Korea | In vitro | ND | Anti-oxidant activity |

Notes.

A549 cell, human lung carcinoma; AMA, antimycin A; ALP, alkaline phosphatase; HaCaT, human keratinocyte cell line; HCT-116, human colon carcinoma; Hela, human cervical carcinoma; HepG2, human hepatocellular carcinoma cells; HUVEC, human umbilical vein endothelial cell; HRECs, primary human retina microvascular endothelial cells; LLC-PK1, (pig kidney epithelium, CL-101); MCF-7, human breast adenocarcinoma; NMDA, $N$-methyl-D-aspartate; ND, not defined; PANC-1, human pancreatic cancer; SNU449, SNU182, human liver cancer cell lines; RAW 264.7, the murine macrophage cell line; VEGF, vascular endothelial growth factor; TPA, 12-$O$-Tetradecanoyl-phorbol-13-acetate; TJ, tight junctions.

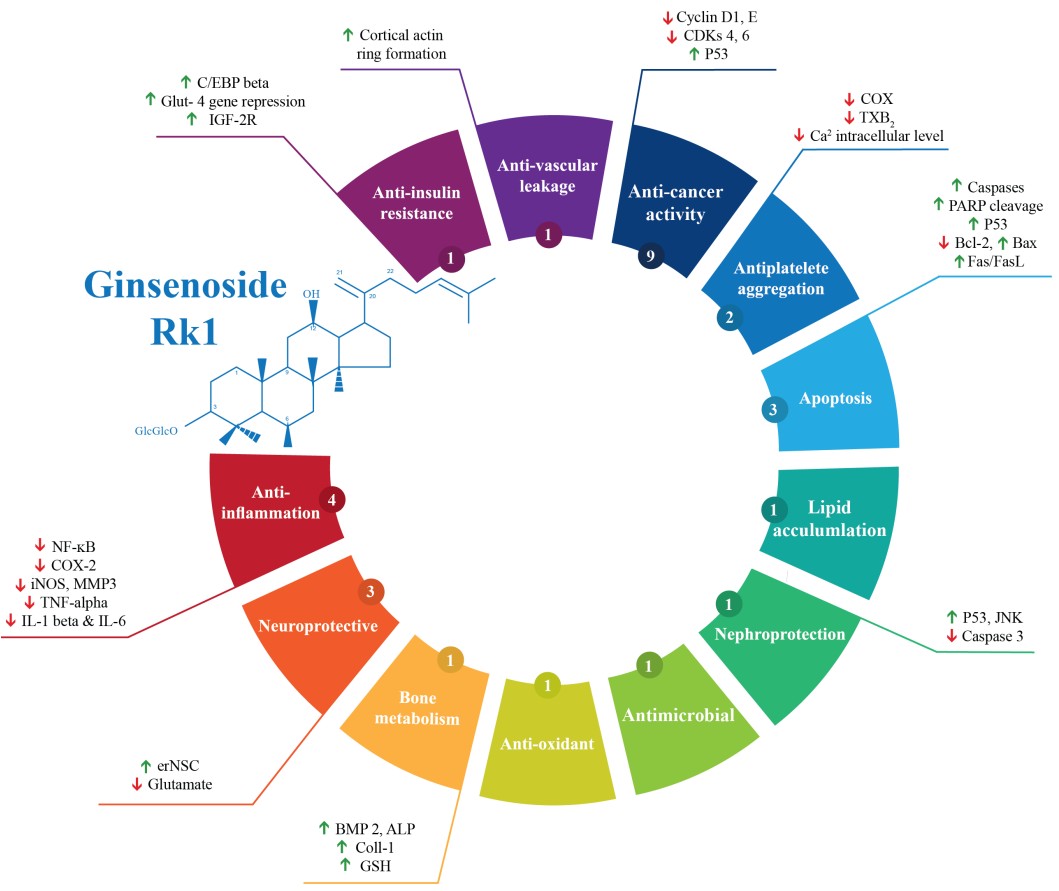

**Figure 3** **Summary of G-Rk1 bioactivities and its mechanism of actions—ALP, alkaline phosphatase; Bax, BCL2-Associated X Protein; Bcl-2, B-cell lymphoma 2; BMP 2, bone morphogenetic protein-2; COX-2, Cyclooxygenase 2; CDK, cyclin-dependent kinase; erNSC, Epidermal growth factor-responsive neurosphere stem cells; GSH, Glutathione; GLUT-4, Glucose Transporter; IL, interleukin; iNOS, Inducible Nitric Oxide Synthase; IGF, insulin-like growth factor receptor; JNK, Jun N-terminal Kinase; MMP3, Matrix Metalloproteinase 3; NF-kB, Nuclear Factor Kappa B; PARP, Poly ADP (Adenosine Diphosphate)-Ribose Polymerase; TXB-2, Thromboxane B2; TNF-α, tumor necrosis factor.** *The number in the small circle indicates the number of studies that report this bioactivity.

## Risk of bias across studies

We used the GRADE method (*Guyatt et al., 2011*) to assess the quality of the included studies. Sixteen studies were categorized as high quality (*Hu et al., 2017*; *Kang et al., 2006*; *Kim et al., 2009*; *Kim et al., 2008*; *Ko et al., 2009*; *Kwak & Pyo, 2016*; *Lee et al., 2009*; *Lee, 2014*; *Lim et al., 2009*; *Liu et al., 2007*; *Maeng et al., 2013*; *Park et al., 2002*; *Quan et al., 2015*; *Ryu et al., 2016*; *Toh et al., 2011*; *Xue et al., 2017*). Twelve studies (*Ahn et al., 2016*; *Bao et al., 2005*; *Jing et al., 2006*; *Ju et al., 2012*; *Kang et al., 2007*; *Kim et al., 2012*; *Kim et al., 2010*; *Kim et al., 2013b*; *Lee et al., 2010*; *Park et al., 2015*; *Ponnuraj et al., 2014*; *Siddiqi et al., 2014*) were categorized as moderate quality. Seven of them (*Ahn et al., 2016*; *Bao et al., 2005*; *Jing et al., 2006*; *Kim et al., 2013b*; *Park et al., 2015*; *Ponnuraj et al., 2014*; *Siddiqi et al., 2014*) focused on the effectiveness of combined (G-Rk1, G-Rg5), thus, they were

downgraded in indirectness item of GRADE factors. Three studies (*Kang et al., 2007*; *Kim et al., 2010*; *Lee et al., 2010*) were not completely pertained to our main outcome since they did not concern mainly with G-Rk1. Four studies (*Kang et al., 2006*; *Lim et al., 2009*; *Liu et al., 2007*; *Park et al., 2002*) were not downgraded in spite of having insufficient data regarding dose-effect factor as this factor does not belong to the downgraded factors of GRADE method that includes (limitation, inconsistency, indirectness, and imprecision). In contrast, one study (*Lee et al., 2010*) that was downgraded since it focused on combined (G-Rk1 and G-Rg5) not because of dose effect insufficient data. Another study (*Kim et al., 2012*) was downgraded because it was used to compare the anti-tumor activity of G-Rk1 versus G-Rk3. However, G-Rk3 has been proven to have a potential antitumor activity. One study (*Ju et al., 2012*) was downgraded as it has statistical typing mistake of one of its values (Table S2).

## Synthesis of results
### Anti-cancer activity
Cell viability was measured by different assays through the studies including four studies used Cell Counting Kit-8 (CCK8) assay (*Kim et al., 2012*; *Kim et al., 2008*; *Kim et al., 2013b*; *Ko et al., 2009*), five studies used 3-(4,5-dimethyl-thiazol-2yl) -2,5-diphenyl tetrazolium bromide (MTT) assay (*Kwak & Pyo, 2016*; *Lee et al., 2010*; *Park et al., 2002*; *Quan et al., 2015*; *Siddiqi et al., 2014*), and one study that used WST-1 assay (*Toh et al., 2011*) while the final one used EZ-CytoTox cell assay kit (*Park et al., 2015*) (Table 2).

*Liver cancer.* *Toh et al. (2011)* evaluated the inhibitory effects of G-Rk1 (0.25 µg/ml) on cell growth of liver cancer cell lines (human hepatocellular carcinoma cells (HepG2), SNU449, and SNU182). A significant reduction of cell viability caused by G-Rk1 at 0.25 mg/ml was recorded ($p < 0.001$). The inhibition concentration ($IC_{50}$) value of G-Rk1 for inhibiting growth in the SNU449 cell line for 48 h was evaluated 0.08 mg/ml (100 µM) by using the WST-1 assay. These results indicated that G-Rk1 is one of the most anti-proliferative ginsenosides of raw and steamed P. notoginseng. Similarly, *Quan et al. (2015)* revealed that the HepG2 cell viability was reduced to 23% and 15% compared to the vehicle control when treated with G-Rk1 at 40 µM and 80 µM for 24 h, respectively (*Quan et al., 2015*).

*Ko et al. (2009)* evaluated the effect of G-Rk1 on cell viability of HepG2 cells after 24 h incubation in concentrations of 50, 75, 100 µM in the presence of 0.1 µM taxol which was used as a positive control. Compared with the vehicle control, G-Rk1 (at a dose of 100 µM) inhibited HepG2 cell proliferation by about 40%. When HepG2 cells were exposed to various concentrations of G-RK1 for 24 h (from 50 to 100 µM), the inhibitory effect on growth rate raised significantly, from 8 to 37.5%, in a dose-dependent manner. In addition, the cell viability was also tested when bafilomycin A1 was added to G-Rk1 (100 µM) and then, three independent experiments showed that this co-treatment enhanced HepG2 cell death more than the cells that were treated with 100 µM of G-Rk1 alone. In this experiment, to verify the effects of this combination and exclude cytotoxicity of bafilomycin A1, cytotoxicity was measured after 24 h and no cytotoxicity was detected.

In the study of *Kim et al. (2008)* they assessed the effects of G-Rk1 on cell viability of HepG2 cells. The concentrations of G-Rk1 ranging from 12.5 to 100 µM with 0.5

Elshafay et al. (2017), *PeerJ*, DOI 10.7717/peerj.3993

**Table 2   Summary of anti-cancer activity of G-Rk1.**

| Author, Year | Cells' origin | Methods/ Cell lines | Methods/ Assays | Methods/ Time effect | Methods/ Treatment | | | | | | Conclusions |
|---|---|---|---|---|---|---|---|---|---|---|---|
| | | | | | Positive control | | | Active compound | | | |
| | | | | | Substance | Concentration | Activity (%) | Substance | Concentration | Activity (%) | |
| **Cell viability inhibition[a]** | | | | | | | | | | | |
| Ko et al. (2009) | South Korea | HepG2 | CCK-8 assay | 24 h | Taxol | 0.1 μM | 100 | G-Rk1 | 50, 75, 100 μM | 92, 70, 62.5 | Inhibition of cell viability in a dose dependent manner |
| Toh et al. (2011) | China | SNU449, SNU182, HepG2 | WST-1. | 48 h (12 h for HepG2) | ND | ND | ND | G-Rk1 | 250 μg/ml | 17.5, 21.1, 18.9 at 24 h for the cell lines respectively | Reduction of cell viability significantly |
| Kim et al. (2008) | South Korea | HepG2 | CCK-8 assay | 48 h | Kit-supplied TSR8 and HeLa cell lysate | ND | ND | G-Rk1 | 25, 50, 75, 100 μM | 89.2, 68.6, 45.7, 5.4 | The first identification of the biological activity of G-Rk1 against HepG2 cell growth |
| Park et al. (2002) | South Korea | SK-Hep-1 cells | MTT assay | ND | ND | ND | ND | G-Rk1 | 5, 10, 20, 50, 100 μM/ml | ND | Inhibition of cell viability |
| Kim et al. (2012) | South Korea | SK-MEL-2 human melanoma | Cell viability assay | 12 h | ND | ND | ND | G-Rk1 | 5, 10, 25, 50, 75, 100 μM | 100, 96, 93.5, 80, 60.5, 18.2 | Inhibition of cell viability of SK-Mel-2 human melanoma cells when they were incubated with G-Rk1 for 24 h and 48 h, at concentrations (0, 10, 25, 50, 75, 100 μM) in a dose dependent manner |
| | | | | 48 h | ND | ND | ND | G-Rk1 | 5, 10, 25, 50, 75, 100 μM | 93.3, 90.5, 81, 65.5, 40.5, 8.8 | |
| Lee et al. (2010) | ND | HUVECs | MTT assay | 24 h | ND | ND | ND | G-Rk1 | 10 μg/ml | 86.29 | Not related to G-Rk1 activity |
| | | | | 48 h | ND | ND | ND | | | 87.42 | |
| Kim et al. (2013b) | South Korea | Gastric cancer AGS cell | CCK-8 assay | 24 h | ND | ND | ND | G-Rg5/G-Rk1 | 0, 12.5, 25, 50, 100 μg/ml | 100, 98.81, 93.57, 93.57, 37.55, 2.94 | Inhibition of cell viability in a dose dependent manner |
| Kwak & Pyo (2016) | South Korea | A549 cell | MTT assay | 24 h | Cisplatin | 5, 10, 25, 50, 100 μM | 84, 73, 72, 51, 37 | G-Rk1 | 5, 10, 25, 50, 100 μM | 87, 83, 83, 73, 18 | Inhibition of cell viability in a dose dependent manner |

| Author, Year | Cells' origin | Methods/ Cell lines | Methods/ Assays | Methods/ Time effect | Positive control | | | Active compound | | | Conclusions |
|---|---|---|---|---|---|---|---|---|---|---|---|
| | | | | | Substance | Concentration | Activity (%) | Substance | Concentration | Activity (%) | |
| Quan et al. (2015) | China | HepG2 cell | MTT assay | 24 h | ND | ND | ND | G-Rk1 | 5, 10, 20, 40, 80 µM | 85, 111, 105, 23, 15 | Reduction of cell viability |
| | | A549 cell | | | | | | | | 97, 93, 110, 47, 3.6 | |
| | | HCT-116 cell | | | | | | | | 99, 103, 70, 6.5, 5.4 | |
| | | Hala cell | | | | | | | | 113, 116, 119, 36, 11 | |
| | | MCF-7 cell | | | | | | | | 125, 127, 123, 84, 8.6 | |
| | | PANC-1 cell | | | | | | | | 98, 115, 104, 24, 9.9 | |
| **Cell proliferation inhibition[b]** | | | | | | | | | | | |
| Ko et al. (2009) | South Korea | HepG2 | CCK-8 assay | 24 h | Taxol | 0.1 µM | 100 | G-Rk1 | 50, 75, 100 µM | 8, 30, 37.5 | G-Rk1 inhibits cell proliferation in the early stage of G-Rk1-induced apoptosis cell line. |
| Toh et al. (2011) | China | SNU449 | WST-1. | 48 h | ND | ND | ND | G-Rk1 | 100 µM | 50 | G-Rk1 inhibit cell proliferation in the SNU449 cell line |
| **Apoptotic activity[a]** | | | | | | | | | | | |
| Kim et al. (2008) | South Korea | HepG2 | CCK-8 assay | 48 h | ND | ND | ND | G-Rk1 | 100 µM | ND | G-Rk1 induced an increase in the fraction of early apoptotic cells from 0.46 to 16.23%. |
| Kim et al. (2012) | South Korea | SK-MEL-2 human melanoma | FAS/FASL antagonist analysis | 24 h | Fas/FasL antagonist Kp 7–6 | 1 mM | ND | G-Rk1 | 5, 10, 50, 100 µM | 96, 93.5, 79.5, 18.7 | The apoptotic effect of G-Rk1 might be influenced by other pathways |
| Hu et al. (2017) | China | Mice | Expression of Bax | ND | APAP | 250 mg/kg | 90.7 | G-Rk1 + APAP | 10, 20 mg/kg | 25.5, 39.8 | G-Rk1 has apoptotic effect by increasing Bax expression and decreasing Bcl-2 |
| | | | Expression of Bcl-2 | | | | 12.6 | | | 65.7, 50.5 | |

**Notes.**

APAP, acetaminophen; A549 cell, human lung carcinoma; Bax, BCL2-Associated X Protein; Bcl-2, B-cell lymphoma 2; CCK-8, Cell Counting Kit-8; HaCaT, human keratinocyte cell line; HCT-116, human colon carcinoma; Hela, human cervical carcinoma; HepG2, human hepatocellular carcinoma cells; HUVEC, human umbilical vein endothelial cell; LLC-PK1, (pig kidney epithelium, CL-101); MCF-7, human breast adenocarcinoma; MTT, 3-(4,5-dimethyl-thiazol-2yl) -2,5-diphenyl tetrazolium bromide; MC3T3-E1, (RCB1126, an osteoblast-like cell line derived from C57BL/6 mouse calvarias); PANC-1, human pancreatic cancer; SNU449, SNU182, human liver cancer cell lines.
[a] measured by cell viability (%).
[b] measured by cell proliferation inhibition (%).
(v/v) dimethyl sulphoxide added as control and incubated the cells for 48 h were used in this study. At 75 and 100 $\mu$M of G-Rk1, the effect of G-Rk1 induced cell death was maximized to 55% and 95% cell death respectively. In addition, the results revealed that the treatment of HepG2 cells with 100 $\mu$M G-Rk1, the fraction of early apoptotic cells increased from 0.46 to 16.23% and the underlying mechanism by which G-Rk1 induces the mitochondria-independent apoptosis can be through the activation of caspase-8, the signaling cascade of the one not associated with Fas-associated death domain expression.

To increase their cytotoxicity against Sk-Hep-1 hepatoma cancer cells, *Park et al. (2002)* used steamed ginseng which was separated by HPLC and tested with MTT assay to produce many active ginsenosides including G-Rk1. In this study, they found that the isolated G-Rk1 was associated with an inhibitory effect on cell viability in Sk-Hep1 cells. The growth inhibition concentration of G-Rk1 was 13 $\mu$M.

*Lung cancer.* G-Rk1 was evaluated in human lung cancer A549, and cell viability (% to control) was assessed using MTT assay. At the concentration of 50 $\mu$M, there was a statistically significant difference between cisplatin treated cell lines and Rk1 treated cell lines. However, G-Rk1 showed approximately two times higher anticancer activity than Rg5 when treated at 100 $\mu$M. After 24 h treatment, the IC50 values of G-Rk1 and cisplatin were 70, and 50 $\mu$M, respectively. Several proteins were found to be related to the apoptotic effect of G-Rk1 such as calmodulin-like protein, purine nucleoside phosphorylase, adaptor molecular crk, and transaldolase enzyme were increased while biliverdin reductase, aldehyde dehydrogenase, dihydropteridine reductase, and transactive response DNA binding protein-43 were decreased (*Kwak & Pyo, 2016*). In another study, A549 cell viability was reduced to 47% and 3.6% compared to the vehicle control when treated with G-Rk1 at 40 $\mu$M and 80 $\mu$M for 24 h, respectively (*Quan et al., 2015*).

*Melanoma.* To evaluate the inhibitory effect on cell viability of SK-Mel-2 human melanoma cells, *Kim et al. (2012)* incubated these cells with G-Rk1 for 24 and 48 h at different concentrations (0, 10, 25, 50, 75, 100 $\mu$M) in a dose-dependent manner. *Erb et al. (2005)* provoked a controversy with the role of FAS and/or FASL in human malignant melanoma. Therefore, the effect of FAS and/or FASL on cell viability was evaluated by *Kim et al. (2012)* by adding Fas/FasL antagonist Kp 7–6 of concentration 1 mM and incubated it for 1 h. Then, the cells were treated with various concentrations of G-Rk1 (1, 5, 10, 50 and 100 $\mu$M). The results showed that Kp 7–6 treatment alone did not induce cell death or cell proliferation. Therefore, they concluded that Kp 7–6 has no effect on cell viability when used alone. However, when the cells were treated with Kp 7–6 followed by G-Rk1 (100 $\mu$M) treatment, the effect of G-Rk1 was reduced by 32 % compared to the control (no treatment of Kp 7–6). Moreover, they also assessed the induction of apoptosis by G-Rk1 in SK-MEL-2-Human Melanoma and their findings showed that when the concentration of G-Rk1 increased, the number of apoptotic cells also increased. More importantly, the cell lines responded in a dose-dependent manner.

*Other types of cancer.* *Kim et al. (2013b)* evaluated the effect of the combination of G-Rg5/G-Rk1 on cell viability of gastric cancer cells. After treatment with this combination at
different concentrations (12.5, 25, 50 and 100 µM) for 24 h, the results showed an inhibitory effect on cell viability and proliferation of these cells in a dose-dependent manner (99, 93.5, 37.5, 3 %) respectively. In another study, cell viability was assessed using different cancer cell lines including human colon carcinoma (HCT-116), human cervical carcinoma (Hela), human breast adenocarcinoma (MCF-7), and human pancreatic cancer (PANC-1). When they were treated with 80 µM of G-Rk1, cell viability was reduced by 5.4%, 11%, 8.6%, and 9.9%, respectively (*Quan et al., 2015*).

### Antiplatelet aggregation activity

Two studies evaluated the anti-aggregation effects of G-Rk1 both *in vivo* and *in vitro* (*Ju et al., 2012*; *Lee et al., 2009*) respectively. *Ju et al. (2012)* compared the antiplatelet aggregation activity of G-Rk1 and acetylsalicylic acid (ASA). The results indicated that G-Rk1 exhibits a stronger antiplatelet aggregation activity than ASA in which the action of G-Rk1 in platelets might be related to arachidonic acid (AA) metabolism. In addition, the alteration of (S) hydroxyl eicosatetraenoic acids and thromboxane B2 levels were determined using an immunoassay kit and UPLC/Q-TOF MS system, respectively. The 12-hydroxyleicosatetraenoic acid level was remarkably decreased in the G-Rk1 group but increased in the ASA-treated group. The thromboxane B2 level in the washed platelets decreased significantly by 66% when treated with 100 µM ASA and 77% when treated with 10 µM G-Rk1 (*Ju et al., 2012*). They used the colorimetric COX inhibitor screening assay to measure the inhibitory effects of G-Rk1 on COX-1 and COX-2. It was found that G-Rk1 inhibits both COX-1 and COX-2 activities. However, at a concentration of 20 µM, G-Rk1-derived inhibition was higher on COX-2 than on COX-1 (*Ju et al., 2012*).

*Lee et al. (2009)* explained in his study that the effect of G-Rk1 on adenosine diphosphate (3–4 µM) induced platelet aggregation was monitored turbidimetrically by using ASA as a positive control. Both ASA and G-Rk1 showed the dose-dependent inhibitory effect on collagen, AA, and U46619 (9,11-dideoxy-11a,9a-epoxymethanoprostaglandin F2a) (thromboxane A2 mimetic drug)-induced platelet aggregation. However, they showed a negligible effect on adenosine diphosphate-induced aggregation. G-Rk1 exhibited the strongest inhibitory effect on collagen, AA, and U46619-induced platelet aggregation. In particular, it presented a 22-fold activity of ASA on AA-induced aggregation (*Lee et al., 2009*). G-Rk1 was found to be a potent inhibitor of AA and U46619 -induced platelet aggregation (Table 3).

### Anti-inflammatory activity

G-Rk1 was found to have an anti-inflammatory effect by inhibiting NF-κB levels in the *in vitro* models (*Lee, 2014*). These results were assessed using luciferase assay. HepG2 cells were seeded at $1 \times 10^5$ cells/well in a 12-well plate and grown for 24 h. While G-Rk1 was pretreated with dimethyl sulphoxide for 1 h and then it was treated with tumor necrosis factor-α (10 ng/mL), the sulfasalazine was used as positive control. Their data demonstrated the strong inhibitory activity of G-Rk1 on NF-κB expression with 50% (IC50) value from 0.75 µM. However, the results revealed that G-Rk1 had cytotoxic effects, which occur in concentrations higher than 10 µM. Another evaluation of G-Rk1 anti-inflammatory activity (*Kim et al., 2010*) was its suppressing effect on 12-*O*-tetradecanoyl- phorbol-13-acetate
**Table 3  Summary of the effects of G-Rk1 on antiplatelet aggregation, anti-inflammatory, anti-vascular leakage, nephroprotective effect, neuroprotective effect, bone metabolism, anti-insulin resistance effect, and lipid accumulation.**

| Author, Year | Cells' origin | Methods/ Cell lines | Methods/ Assays | Methods/ Time effect | Methods/ Treatment | | | | | | Conclusions |
|---|---|---|---|---|---|---|---|---|---|---|---|
| | | | | | Positive control | | | Active compound | | | |
| | | | | | Substance | Concentration | Activity | Substance | Concentration | Activity | |
| **Antiplatelet aggregation** | | | | | | | | | | | |
| *Ju et al. (2012)* | South Korea | Platelet | A UPLC/Q-TOF MS system | ND | ASA | 50 $\mu$M | ND | G-Rk1 | 50 $\mu$M | ND | G-Rk1 strongly inhibited platelet aggregation at 50 $\mu$M compared with ASA |
| *Lee et al. (2009)* | South Korea | Platelet | Turbidimetrically | ND | ASA | 66 $\mu$M | 50 | G-Rk1 | 3 $\mu$M | 50[a] | G-Rk1 exhibited 22-fold inhibitory effect of that of ASA on AA-induced aggregation |
| **Anti-inflammatory activity** | | | | | | | | | | | |
| *Lee (2014)* | South Korea | HepG2 | NF-$\kappa$B-luciferase assay | 1 h | Sulfasalazine | 0.54 $\mu$M | 50 | G-Rk1 | 0.75 $\mu$M | 50[a] | G-Rk1 exhibited the potentials as anti-inflammatory substance against hepatitis |
| *Kim et al. (2010)* | South Korea | Collagen-induced mouse arthritis model | Edema | 4 h | ND | ND | ND | G-Rk1 | 10, 50 mg/kg | 9.09, 7.83[b] | G-Rk1 exhibited anti-inflammatory activity on collagen-induced mouse arthritis model |
| *Ahn et al. (2016)* | South Korea | HaCaT/RAW 264.7 | TARC/CCL17 | 1 h | TNF-$\alpha$ | 10 ng/mL | 157 pg/ml | G-Rg5/G-Rk1 + TNF-$\alpha$ | 1, 25, 50 $\mu$g/ml | 118, 104, 95.4 pg/ml | The results suggesting G-Rg5/G-Rk1 as a promising natural therapy in the control of atopic dermatitis |
| | | | MDC/CCL22 | | | | 243 pg/ml | | | 215, 209, 189 pg/ml | |
| *Hu et al. (2017)* | China | | Expression of TNF-$\alpha$ | 1 h | APAP | 250 mg/kg | 156 ng/L | G-Rk1 + APAP | 10, 20 mg/kg | 87, 96.7 ng/L | G-Rk1 has a protective effect against APAP induced liver injury in mice by decreasing the expression of TNF-$\alpha$ and IL-1B |
| | | | Expression of IL-1B | | | | 1,550 pg/L | | | 1,060, 1,140 pg/L | |

Elshafay et al. (2017), *PeerJ*, DOI 10.7717/peerj.3993

**Table 3** (*continued*)

| Author, Year | Cells' origin | Methods/ Cell lines | Methods/ Assays | Methods/ Time effect | Positive control | | | Active compound | | | Conclusions |
|---|---|---|---|---|---|---|---|---|---|---|---|
| | | | | | Substance | Concentration | Activity | Substance | Concentration | Activity | |
| **Anti-vascular leakage** | | | | | | | | | | | |
| *Maeng et al. (2013)* | South Korea | HRECs | Sucrose permeability assay | 1 h | ND | ND | ND | G-Rk1 | 10 μg/ml | 114.72[c] | G-Rk1 exhibited an inhibitory effect of VEGF-induced vascular permeability in the mouse retina |
| **Effect of G-Rk1 on lipid accumulation** | | | | | | | | | | | |
| *Kim et al. (2009)* | South Korea | 3T3-L1 fibroblast cells | Oil red O staining | 2 h | ND | ND | ND | G-Rk1 | 10, 50, 100 | 0.11, 0.12, 0.08[d] | G-Rk1 showed inhibitory effect on lipid accumulation in 3T3-L1 adipocytes |
| **Nephroprotective effect** | | | | | | | | | | | |
| *Park et al. (2015)* | South Korea | LLC-PK1 cells | EZ-Cytox cell viability assay kit | 24 h | EGCG (without cisplatin) | 0 μg/ml | 99.58% | G-Rg5/G-Rk1 | | 100.0% | G-Rg5 and G-Rk1 showed a protective effect against cisplatin-induced nephrotoxicity in cultured kidney cells and mice |
| | | | | | EGCG | 0, 50, 100, 250 μg/ml | 40.76, 46.34, 47.33, 38.65% | G-Rg5/G-Rk1 (cisplatin 25 uM) | 0, 50, 100, 250 μg/ml | 40.23, 45.23, 57.32, 80.21% | |
| **Bone metabolism** | | | | | | | | | | | |
| *Siddiqi et al. (2014)* | South Korea | The murine cell line, MC3T3-E1 | MTT assay | 24 h + 48 h | AMA | 60 μg/ml | 99.33% | G-Rg5/G-G-Rk1 + AMA | 1, 10, 20, 30, 50 μg/ml | 109.21, 111.54, 123.43, 131.21, 140.05% | G-Rg5/G-Rk1 enhances cell growth of MC3T3-E1 cells in a dose-dependent manner, also in presence of AMA |
| **Neuroprotective effect** | | | | | | | | | | | |
| *Bao et al. (2005)* | South Korea | Mice | ND | ND | Ethanol | 3 g/kg | 34.5, 44.22[e] | G-Rg5/G-G-Rk1 | 10 mg/kg | 45.68, 207.48[e] | G-Rg5/G-G-Rk1 significantly reversed the memory dysfunction that was induced by ethanol or scopolamine |
| *Jing et al. (2006)* | China | Mice | ND | ND | Ethanol | ND | 35, 62[e] | G-Rg5/G-G-Rk1 | 2, 10 mg/kg | 44, 50/ 145, 184[e] | The results suggest that those compounds have the ability to improve the acquisition of ethanol-treated mice |
**Table 3** (*continued*)

| Author, Year | Cells' origin | Methods/ Cell lines | Methods/ Assays | Methods/ Time effect | Methods/ Treatment | | | | | | Conclusions |
|---|---|---|---|---|---|---|---|---|---|---|---|
| | | | | | Positive control | | | Active compound | | | |
| | | | | | Substance | Concentration | Activity | Substance | Concentration | Activity | |
| **Anti-insulin resistance effect** | | | | | | | | | | | |
| *Ponnuraj et al. (2014)* | South Korea | 3T3-L1 | MTT assay | 24 h | ND | ND | 1.36 | G-Rg5/G-Rk1 | 25, 50, 75, 100 $\mu$g/ml | 1.4, 1.36, 1.33, 1.31[f] | G-Rk1 increases the IGF-2R and glucose utilization in adipocytes. |
| | | | | | Tunicamycin | 2 $\mu$g/ml | 0.73 | G-Rg5/G-Rk1 (under Tunicamycin 2 $\mu$g/ml) | | 1.4, 1.47, 1.5, 1.54[f] | |

**Notes.**

AA, arachidonic acid; AMA, antimycin A; ASA, acetylsalicylic acid; APAP, acetaminophen; A549 cell, human lung carcinoma; Bax, BCL2-Associated X Protein; Bcl-2, B-cell lymphoma 2; CCK-8, Cell Counting Kit-8; EGCG, Epigallocatechin gallate; HaCaT, human keratinocyte cell line; HCT-116, human colon carcinoma; Hela, human cervical carcinoma; HepG2, human hepatocellular carcinoma cells; HRECs, Primary human retina microvascular endothelial cells; IL-1$\beta$, interleukin-1$\beta$; LLC-PK1, (pig kidney epithelium, CL-101); MCF-7, human breast adenocarcinoma; MDC/CCL22, macrophage-derived chemokine; MTT, 3-(4,5-dimethyl-thiazol-2yl) -2,5-diphenyl tetrazolium bromide; MC3T3-E1, (RCB1126, an osteoblast-like cell line derived from C57BL/6 mouse calvarias); PANC-1, human pancreatic cancer; SNU449, SNU182, human liver cancer cell lines; TNF-$\alpha$, tumor necrosis factor-alpha; TARC/CCL17, thymus and activation-regulated chemokine.

[a]50% inhibition concentration (IC$_{50}$) values.
[b]was indicated as the increase in weight of the right ear punch over that of the left (mg).
[c][$^3$H] sucrose permeability (%).
[d]measured by the optical absorbance at 490 nm.
[e]Latency by seconds for learning and testing respectively.
[f]Cell viability was measured based on absorbance values at 570 and 630 nm.

(TPA) induced mouse ear edema. The right ear of ICR mouse was treated with red ginseng saponin extract, G-Rg3, G-Rg5, and G-Rk1 of 10, and 50 mg/kg and after 30 min, ear edema in both ears was induced by topical application of TPA, which is a potent inflammatory agent. They measured the extent of edema and noticed that the pretreatment with red ginseng saponin extract or G-Rk1 suppresses TPA-induced mouse ear edema, and when administering G-Rk1 orally, the formation of edema was blocked. *Hu et al. (2017)* showed that in acetaminophen (APAP) induced liver injury in mice, G-Rk1 can be used as a protective agent, as it significantly reduced the levels of tumor necrosis factor (TNF-$\alpha$) to 87 ng/L and when treated with 10 mg/kg G-Rk1 compared to 156 ng/L when treated with 250 mg/kg APAP. A significant reduction of interleukin-1$\beta$ (IL-1$\beta$) was observed with G-Rk1 (*Hu et al., 2017*). Atopic dermatitis in which keratinocytes and macrophages produce excess chemokines and cytokines, especially thymus and activation-regulated chemokine (TARC/CCL17) and macrophage-derived chemokine (MDC/CCL22), as well as nitric oxide (NO), Ahn's results using G-Rg5/G-Rk1 on TNF-$\alpha$/ IFN-$\gamma$ stimulated human keratinocytes cell line (HaCaT cells) showed a significant reduction of TARC/CCL17 expression. Furthermore, using the same combination on the murine macrophage cell line RAW264.7, the secretion of lipopolysaccharide (LPS) mediated NO and reactive oxygen species were significantly reduced, suggesting G-Rg5/G-Rk1 as a promising natural therapy in the control of atopic dermatitis (*Ahn et al., 2016*) (Table 3).

### Effect of G-Rk1 on vascular leakage

A study evaluated the G-Rk1 effect on VEGF (*Maeng et al., 2013*) by treating primary human retina microvascular endothelial cells with G-Rk1 at a concentration of (10 $\mu$g/ml) for 40 min then stimulating it with 20 $\mu$g/ml of VEGF to disrupt the cell membrane. Sucrose permeability assay was used to evaluate the endothelial permeability and the results showed that G-Rk1 inhibited VEGF-induced retinal endothelial permeability. They used reverse-transcription polymerase chain reaction (RT-PCR) and densitometric analysis was used to assess translocation of tight junctions (TJ) proteins, and immunostaining was used to evaluate disruption of TJ proteins after the cells were stained with anti-ZO-1, anti-ZO-2, and anti-occludin antibodies. The authors found that G-Rk1 inhibited VEGF effect on TJ protein localization but it did not affect the transcription of TJ proteins (Table 3).

### Effect of G-Rk1 on lipid accumulation

Ginseng is known to have effects on obesity (*Kim et al., 2009*). *In vitro* treatment of mouse 3T3-L1 fibroblast cells with G-Rk1 resulted in reducing lipid accumulation, in which these cells differentiated into adipocytes after being treated with various G-Rk1 concentrations (10, 50, 100 $\mu$M) for 2 h at 490 nm optical absorbance (*Kim et al., 2009*) (Table 3).

### Neuroprotective effect of G-Rk1

The combination of G-Rg5/G-Rk1 had a pronounced effect on the excitotoxic and oxidative stress-induced neuronal cell damage that was tested in primary cultured rat cortical cells (*Bao et al., 2005*). These cells were cultured *in vitro* for 12–20 days, then exposed to 100 $\mu$M glutamate or N-methyl-D-aspartate for 15 min in the absence or presence of

G-Rg5/G-Rk1. The cell damage was assessed after 20–24 h by measuring LDH activity in the culture media. Data were calculated from cells exposed to the respective excitotoxic insults without ginsenosides. Data presented that approximately 70–80% of the cells were damaged by glutamate or N-methyl-D-aspartate compared to vehicle-treated control cells. The excitotoxic effect was significantly inhibited by G-Rg5/G-Rk1 in a concentration-dependent manner, in which 50% inhibition was achieved at 14.7 μg/mL of G-Rg5/G-Rk1.

In previous work, *Bao et al. (2005)* used a passive avoidance test to evaluate the effect of G-Rg5/G-Rk1. The latency in seconds was used to measure the cognitive performance of ethanol-induced amnesia in mice. The mice were orally treated with saline as vehicle and the ratio of G-Rg5/G-Rk1 equal 1:1 with a concentration of 10 mg/kg once a day for 4 days. The latency period of the mice administrated with ethanol was 24.9% less than the one of control mice (without ethanol-treatment), but it was significantly enhanced by the oral administration of G-Rg5/G-Rk1 with 1.2-fold increase than that of the control. The same steps were done, but this time after inducing amnesia with a single injection of scopolamine (3 mg/kg), also G-Rg5/G-Rk1 (10 mg/kg) provided the same enhancing significant result ($p < 0.01$). In another work, *Jing et al. (2006)* did the same tests of ethanol-induced amnesia in mice, which were given water as the control and ratio of G-Rg5/G-Rk1 equal 1:1 in the concentration of 10 mg/kg. They found that G-Rg5/G-Rk1 could significantly prolong the latency period by 2.97 folds more than that of the control. These two studies presented that G-Rg5/G-Rk1 would give beneficial results in the memory function of the normal, ethanol or scopolamine-induced amnesia in brains. G-Rk1 was reported to have a significant neurogenic activity in Epidermal growth factor-responsive neurosphere stem cells (erNSCs). However, this activity was less than G-Rg5 (*Liu et al., 2007*) (Table 3).

### Nephroprotective effect of G-Rk1

*Park et al. (2015)* examined the effect of the G-Rg5/G-Rk1 combination on cisplatin-induced nephrotoxicity in mice at cisplatin concentration 25 μM and G-Rg5/G-Rk1 concentrations of (0, 50, 100, 250 μg/ml). Results with EZ-cytotoxic cell viability assay kit showed a significant reduction in cisplatin and induced a reduction in cell viability. This effect was higher than that of Epigallocatechin gallate at the same concentrations as G-Rk1 (Table 3).

### Bone metabolism

*Siddiqi et al. (2014)* evaluated the osteogenic activity of G-Rg5/G-Rk1. MC3T3-E1 cells were treated with differentiation medium (either with or without G-Rg5/G-Rk1) for 12 days at different concentrations in which different substances were added to the culture medium in order to evaluate various effects of G-Rg5/G-Rk1 on differentiated fibroblast. The extent of calcium deposition, which is an indicator of osteoblasts mineralization, was measured by MTT assay. Data were expressed as a percentage of control, which showed that G-Rg5/G-Rk1 protected the extracellular matrix mineralization from antimycin A devastating effects. Besides, it turned out that alkaline phosphatase (ALP) activity evaluated by Smart BCA protein assay kit, increased by two folds after treatment with G-Rg5/G-Rk1 (30–50 μg/mL).

The effect of G-Rg5/G-Rk1 on cellular collagen was measured using Sirius Red-based colorimetric assay. Results were similar to that of ALP activity in which cellular collagen was markedly increased. When glutathione contents of the cells were measured by glutathione assay kit after exposure to various concentrations of G-Rg5/G-Rk1, data showed that G-Rg5/G-Rk1 increase the level of glutathione in a dose-dependent manner. In order to evaluate gene expression levels, a total RNA was isolated from the cells, which were treated with G-Rg5/G-Rk1 and was amplified by RT-PCR. The results indicated that the maturation and the differentiation of MC3T3-E1 cells were induced by G-Rg5/G-Rk1 mediated BMP-2/Runx2 and the level of expression of Runx2 increased by the action of G-Rg5/G-Rk1 (Table 3).

### Anti-insulin resistance effect of G-Rk1

*Ponnuraj et al. (2014)* assessed the effect G-Rk1 on insulin resistance. 3T3-L1 cells were treated with G-Rg5/G-Rk1 complex at different concentrations where tunicamycin was used to induce stress on the endoplasmic reticulum (ER). As for cell viability, measured with MTT assay, results showed that cells treated with G-Rg5/G-Rk1 complex had overcome the stress which induced by tunicamycin. Cells were made insulin resistant by immersing them into a medium that contains insulin and by treating them with dexamethasone, then with the stress agent and G-Rg5/G-Rk1 complex and were analyzed by glucose oxidase reagent, while tunicamycin was used as a positive control. Results found that the amount of glucose left in the medium is high in the cells treated with tunicamycin and low in the cells treated with G-Rg5/G-Rk1 complex and this was achieved through C/EBP homologous protein-10 (CHOP)-mediated pathway and increase insulin-like growth factor receptor (IGF-2R) (Table 3).

### Anti-oxidant effect of G-Rk1

Hydrothermal treatment of primary ginsenosides at 100 C transformed them into either deglycosylated and/or dehydrated ginsenoside. As the hypothermal reaction increase, it yields more 20 (S)-Rg3, Rk1, and Rg5. In addition, when they compared the antioxidant activity between the hydrothermally processed samples at 100 C and others processed by the steaming, they got results that showed that sun ginseng samples were higher in antioxidant activities. However, it results in fewer ginsenosides than these samples which reacted at 120 C (*Ryu et al., 2016*).

### Antimicrobial effect of G-Rk1

*Xue et al. (2017)* assessed the antimicrobial effect of G-Rk1 measured by the minimum inhibitory concentration (MIC) and minimum bacterial concentration (MBC). Compared to Erythrocin (positive control), G-Rk1 exhibited higher MIC and MBC against different bacterial strains compared to Erythrocin (positive control) with (MIC: 31.3 vs 8 $\mu$g/ml; MBC: 125.0 vs 16.0 $\mu$g/ml) against *Clostridium perfringens*, (MIC: 16.0 vs 8 $\mu$g/ml; MBC: 125.0 vs 31.3 $\mu$g/ml) against *Fusobacterium nucleatum, and* (MIC: 62.5 vs 16.0 $\mu$g/ml; MBC: 125.0 vs 62.5 $\mu$g/ml) against *Porphyromonas gingivalis.* Therefore, G-Rk1 can be a promising cure for halitosis.

## DISCUSSIONS

Ginsenosides are active compounds extracted from white or red ginseng (P. Ginseng Meyer). Ginsenosides have shown pharmacological effects on the cardiovascular system (*Sun, Liu & Chen, 2016*), the immune system (*Song, Zang & Hu, 2009*), and the central nervous system (*Zhou et al., 2014*), as well as anti-stress, antioxidant, and anti-cancer activities. Moreover, ginsenosides have shown good results in the treatment of diabetes disease by improving glucose and insulin control in type 2 diabetes in a clinical trial (*Vuksan et al., 2008*). Antitumor inhibitory effects of ginsenosides have been demonstrated because of their cytotoxicities such as the suppression of tumor angiogenesis and metastasis by G-Rb2 (*Sato et al., 1994*) and the enhancement of apoptosis by G-Rg3 in various cancer cell lines such as breast cancer (*Kim et al., 2013a*). Although G-Rk1 has a similar structure to G-Rg3, G-Rk1 could be formed by processing ginseng at high temperature, but its antitumor activities would have been limited. Its pharmacological activity has been assessed on antitumor activity in human hepatocellular carcinoma cells (*Kim et al., 2008*). Apart from these activities, G-Rk1 has been demonstrated to ameliorate impaired memory function and prevent platelet aggregation (*Lee et al., 2009*). Furthermore, G-Rg3, G-Rk1, and G-Rg5 exhibited a potential effect in the management of human arthritis (*Kim et al., 2010*).

In this systematic review, we found that various pharmacological and therapeutic effects of G-Rk1 have been reported in the 28 included studies such as anti-cancer effects (*Kim et al., 2008*), antiplatelet aggregation activities (*Ju et al., 2012*; *Lee et al., 2009*), cognitive function enhancement (*Bao et al., 2005*), anti-inflammatory effects (*Kim et al., 2010*; *Lee, 2014*), lipid accumulation reduction (*Kim et al., 2009*), antioxidant effects (*Ryu et al., 2016*), anti-insulin resistance (*Ponnuraj et al., 2014*), and protection against human arthritis and nephrotoxicity (*Kim et al., 2010*; *Park et al., 2015*).

Anti-cancer activity is one of the most common bioactivities of G-Rk1. By assessing such studies in "cell viability"/"cell proliferation inhibition" and "apoptotic activity", these studies exhibited the anti-cancer effects of G-Rk1 in *in vitro* studies as well as the combined effect of G-Rg5/G-Rk1 (ratio equal 1:1). In terms of "cell viability", the effects of G-Rk1 on cell viability of HepG2 cells, SNU449, SNU182, SK-Hep-1, SK-Mel-2, Hela, HCT-116, MCF-7, PANC-1, A549, and human malignant melanoma was found significantly in a dose-dependent manner (*Kim et al., 2012*; *Kim et al., 2008*; *Ko et al., 2009*; *Kwak & Pyo, 2016*; *Quan et al., 2015*). The concentrations of G-Rk1 vary from 0 to 100 $\mu$M, and the cytotoxic effect was maximum at 75 and 100 $\mu$M (*Kim et al., 2008*). The effects of G-Rk1 were also evaluated in combination with other chemotherapeutics (Bafilomycin A1)(*Ko et al., 2009*).

It was found that the enhancement of HepG2 cell death was higher when applying G-Rg5/G-Rk1 combination than that of G Rk1 alone. Furthermore, we also found three relevant studies (*Kim et al., 2013b*; *Park et al., 2015*; *Siddiqi et al., 2014*) that evaluated the effects of G-Rg5/G-Rk1 co-treatment on cell viability of gastric cancer cells, mice, and MC3T3-E1 cells. The authors demonstrated that G-Rg5/G-Rk1 has potential effects on inhibiting cell viability and proliferation in a dose-dependent manner. The combination of G-Rg5/G-Rk1 with others chemotherapies (cisplatin (*Park et al., 2015*), antimycin

A (*Siddiqi et al., 2014*)) has a greater effect on cell death than using G-Rg5 or G-Rk1 alone. Besides, it was also proved that co-administration of G-Rg5/G-Rk1 with a ratio 1:1 have various effects such as improving the cognitive performance in ethanol-induced amnesia in mice (*Bao et al., 2005*; *Jing et al., 2006*), inhibiting the exotoxic and oxidative stress-induced neuronal cell damage (*Bao et al., 2005*), and stimulating the mineralization of the extracellular matrix of osteoblasts (*Siddiqi et al., 2014*).

In this systematic review, we found two studies presenting the antiplatelet aggregation activities with the results indicating that G-Rk1 ($10 \, \mu M$) can be stronger than ASA ($100 \, \mu M$) regarding the antiplatelet aggregation (*Ju et al., 2012*). *Lee et al. (2009)* also showed that G-Rk1 inhibited the effects of collagen, AA, and U46619-induced platelet aggregation. G-Rk1 was also indicated as one of the effective anti-inflammatory agents through the inhibition of both COX1 and COX2 activities and NF-κB levels (*Ju et al., 2012*; *Lee, 2014*).

Although, more than ten of our included studies reported that G-Rk1 has an anti-cancer effect against different cancer cell lines, all of them were *in vitro* studies with no *in vivo* or clinical studies. Unlikely, it was reported that G-Rg3 has an anti-cancer effect in both *in vitro* and *in vivo* (*Shan et al., 2014*). A recent meta-analysis of randomized clinical trials revealed that G-Rg3 combined with chemotherapy for non-small-cell lung cancer could enhance the overall survival rate and alleviate the chemotherapy-induced side effects (*Xu et al., 2016*). The shortage of *in vivo* or clinical studies to assess the G-Rk1 anti-cancer effect may raise many questions regarding the effect of G-Rk1 in patients and whether it differs from its *in vitro* action. In addition, what alterations that may occur in the patients. Therefore, there is a need for *in vivo* experiments to confirm the G-Rk1 anti-cancer activity and its mechanism.

Regarding the methodological approaches, several limitations were encountered. One of them is that we could not find any clinical study that used G-Rk1 in patients or healthy people. Out of 317 studies, we included 28 studies using our criteria, they were *in vitro* studies and *in vivo* animals. Based on the GRADE method, seven studies remained because of indirectness of evidence (*Ahn et al., 2016*; *Bao et al., 2005*; *Jing et al., 2006*; *Kim et al., 2013b*; *Park et al., 2015*; *Ponnuraj et al., 2014*; *Siddiqi et al., 2014*) and inability to explain heterogeneity in results (*Bao et al., 2005*). To date, there is a shortage of literature regarding clinical studies and the clinical use of G-Rk1 to treat some diseases in patients, and it consequently prohibits the clinical analysis.

## CONCLUSIONS

In general, G-Rk1 has a significant anti-tumor effect on liver cancer, melanoma, lung cancer, cervical cancer, colon cancer, pancreatic cancer, gastric cancer, and breast adenocarcinoma against *in vitro* cell lines. Furthermore, *In vivo* experiments are necessary to confirm these effects. Additionally, G-Rk1 has demonstrated several pharmacological effects such as antiplatelet aggregation, anti-inflammatory, anti-oxidant, antimicrobial, anti-insulin resistance, neuroprotective, nephroprotective, and anti-lipid accumulation effects. All of these results support the clinical effects of G-Rk1 and demonstrate the promising possibility to develop the G-Rk1-based treatments, either alone or in combination with G-Rg5, for the previously mentioned conditions.

## ACKNOWLEDGEMENTS

The authors wish to thank Eman Al Sagheir, (Faculty of Medicine, Zawia University, Libya) for her efforts to revise and improve the English of our manuscript.

### Funding

This study was supported in part by a "Grant-in-Aid for Scientific Research (B)" (16H05844, 2016–2019 for Nguyen Tien Huy) from Ministry of Education, Culture, Sports, Science, and Technology (MEXT) of Japan and by the Japan Initiative for Global Research Network on Infectious Diseases (J-GRID) for Kenji Hirayama. The funders had no role in study design, data collection, and analysis, decision to publish, or preparation of the manuscript.

### Grant Disclosures

The following grant information was disclosed by the authors:
Ministry of Education, Culture, Sports, Science and Technology (MEXT) of Japan: 16H05844, 2016–2019.
Japan Initiative for Global Research Network on Infectious Diseases (J-GRID).

### Competing Interests

Authors Abdelrahman Elshafay, Ngo Xuan Tinh, Samar Salman, Yara Saber Shaheen, Eman Bashir Othman, Mohamed Tamer Elhady, Aswin Ratna Kansakar, Linh Tran, and Le Van are members of Online Research Club (ORC). Nguyen Tien Huy is the founder of ORC.

### Author Contributions

- Abdelrahman Elshafay, Ngo Xuan Tinh, Samar Salman, Yara Saber Shaheen and Eman Bashir Othman conceived and designed the experiments, performed the experiments, analyzed the data, contributed reagents/materials/analysis tools, wrote the paper, prepared figures and/or tables, reviewed drafts of the paper.
- Mohamed Tamer Elhady, Aswin Ratna Kansakar and Linh Tran conceived and designed the experiments, performed the experiments, analyzed the data, contributed reagents/materials/analysis tools, wrote the paper, reviewed drafts of the paper.
- Le Van, Kenji Hirayama and Nguyen Tien Huy conceived and designed the experiments, performed the experiments, contributed reagents/materials/analysis tools, wrote the paper, reviewed drafts of the paper.

### Data Availability

The raw data is included in Tables 1–3.

## Supplemental Information

Supplemental information for this article can be found online at http://dx.doi.org/10.7717/peerj.3993#supplemental-information.

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
