# Peer review of "Ginsenoside Rk1 bioactivity: a systematic review"

_PeerJ, doi:10.7717/peerj.3993_

## Round 0.1 · original submission · Major Revisions

Please address all the comments of the reviewers.

Specifically, you may need comprehensive English editing and to incorporate recent works on the topic. As one of the reviewers pointed out, the current manuscript only include works until 2015.

Reviewer 1 ·

Basic reporting

.

Experimental design

.

Validity of the findings

.

Additional comments

This manuscript handles Ginsenoside Rk1 bioactivity: A systematic review. This is an interesting and well-written manuscript. However, the manuscript contains some weakness that should be revised before it is considered for the journal publication:

Line 66: Genus Panax can be classified by origins. Chinese ginseng is Panax notoginseng and Korean ginseng is Panax ginseng “traditional Chinese medicines” should be changed or rephrased to “traditional medicines”.

Comparison with other ginsenosides will be needed.

Reviewer 2 ·

Basic reporting

no comment

Experimental design

no comment

Validity of the findings

no comment

Additional comments

Authors curated 21 original research papers showing biological and pharmacological effects of ginsenoside Rk1 and summarized findings of each paper in the manuscript. Due to the style that seems to be common in medical fields, authors carefully enlist endless effects of Rk1 on endless lists of disease without providing any conceptual framework on how Rk1 works. This could be due to original papers that hardly provide any mechanism. Ideally, authors should have tried to make a sense out of all those different effects by extrapolating signaling mechanisms behind all those diseases. Nevertheless, the manuscript is resourceful and should help readers who want to know reported effects of Rk1 on various disease.

Reviewer 3 ·

Basic reporting

1) The manuscript needs to be checked and re-edited for grammatical and punctuation errors. I would suggest that the authors revise the abstract, as lines 51-58 seem repetitive in nature.
2) The manuscript summarizes all relevant papers and provides a good overview of the literature but this study has only been performed till Aug 2015 and not till the present day (in this case May/June 2017). Thus, it is recommended to revise this manuscript to include all the literature that has been reported till present date.
3) How many different types of ginsenosides are there in total? It would be better to include a brief description along with a figure in the introduction, which talks about all the different ginsenosides, and how they are isolated. This will help the readers understand better, how Ginsenoside Rk1 is an entirely different class of compound and exhibits unique bioactivity when compared with its sister analogues.
4) The results section on the anti-cancer activity including sub-titles liver cancer, melanoma and gastric cancer is very descriptive, repetitive and needs to be re-edited. The authors have already provided similar information in table 2. I recommend thoroughly revising this particular part of the manuscript to reflect critical findings. Avoid using the word “they” excessively within the anti-cancer section. The other half of the bioactivity section (anti-platelet, anti-inflammatory and so on) is well-written.
5) The references are adequately cited within the manuscript
6) Figures and table provide enough relevant information and conforms to the publisher guidelines.

Experimental design

1) The research presented within the manuscript is original and within the scope and aims of the journal.
2) The systemic review or meta-analysis is carried out in a detailed and systematic manner.
3) The GRADE method followed by the authors is a good example of methods that need to be employed to avoid bias in reporting findings. The grading system has also been adequately described by the authors in Supplemental Table 2.

Validity of the findings

The authors tend to be quite positive about the findings published in other articles, without critically assessing related risks and disadvantages. This may in some cases sketch an unrealistic view of Ginsenoside Rk1 bioactivity. For example, the authors describe Rk1 as anti-cancer drug, however, not a whole lot of work has been conducted with this particular compound and majority of the work published has been primarily carried out in vitro. How would the drug behave in vivo when it is administered to tumor bearing mice or cancer patients ? Is the compound stable ? What are the solubility and bioavailability parameters that need to be taken into account to develop them into anti-cancer therapeutics? Such statements would greatly contribute to the novelty and originality of the manuscript, and I strongly recommend revising this in the conclusions/discussions section.

Additional comments

Line 46: Introduce space before we
Line 55: Shouldn’t the word be “on” liver cancer, melanoma and gastric cancer?
Line 71: Re-edit the sentence
Line 128: Change to includes or included rather than including
Line 268: “in the” is repeated twice
Line 272: Remove space between I and t
Line 301: Use “by” instead of “through”
Line 391: “Confirmed the anti-cancer effects” is a strong statement to use considerably due to all the in-vitro studied conducted. Please revise this statement.

---

## Round 0.2 · Minor Revisions

The revised manuscript resolved all the scientific concerns raised by reviewers.

However, the English language needs further improvement before the manuscript can be Accepted. We suggest that you seek English speaking colleagues (or an editing service) to revise the English language, before resubmitting.

---

## Round 0.3 · accepted · Accept

The current version of ms apparently resolved all the concerns raised during review processes.